# When effort pays off: An experimental investigation into action orientation and anxiety as buffering factors between anhedonia and reward motivation

**Michael R. Gallagher**[1,2]*, **Amanda C. Collins**[3,4], **Samantha L. Anduze**[5], **E. Samuel Winer**[5]

**1** Department of Psychology, Mississippi State University, Mississippi State, Mississippi, United States of America, **2** Department of Psychiatry and Behavioral Health, The Ohio State University Wexner Medical Center, Columbus, Ohio, United States of America, **3** Department of Psychiatry, Massachusetts General Hospital, Boston, Massachusetts, United States of America, **4** Department of Psychiatry, Harvard Medical School, Boston, Massachusetts, United States of America, **5** Department of Psychology, The New School for Social Research, New York, New York, United States of America

* mrg463@msstate.edu

## Abstract

Reward motivation, a construct tied to depression, has been studied using the Effort-Expenditure for Rewards Task (EEfRT). Prior work indicates that anhedonia can reduce reward motivation on the EEfRT, as those with higher levels of anhedonia tend to engage in low reward tasks that require less effort as opposed to expending higher levels of effort to obtain a larger reward. Action orientation has shown to act as a buffer at low levels of anhedonia, but this effect has not been seen at high levels of anhedonia. The current study examined if these findings replicated without a stress manipulation and explored the interaction between anxiety and anhedonia in predicting persistence on the EEfRT using two moderation models. Participants (N = 101) with varying levels of depressive symptoms took part in the study. The first model examined the effects of anhedonia and action orientation on reward motivation. The second model investigated the influence of anhedonia and anxiety on reward motivation. Findings revealed that higher levels of anhedonia were significantly associated with lower reward motivation in both models. Additionally, the interaction between anhedonia and action orientation on reward motivation was significant. Trend analyses revealed that, at low levels of anhedonia, participants generally made more high-effort/high-reward choices or were willing to exude more effort for the possibility of obtaining a greater reward. However, as anhedonia increased, individuals with higher levels of action orientation exhibited greater effort as opposed to those with lower action orientation. The findings indicate that anhedonia has a strong impact on limiting reward motivation. However, high levels of action orientation can mitigate the negative influence of anhedonia on reward motivation.

**Data availability statement:** All relevant data are available via the Open Science Framework (OSF; https://osf.io/rv67g/).

**Funding:** Research reported in this publication was supported by the National Institute of Mental Health (NIMH) under Award Number R15MH101573 (PI: ESW). The funders had no role in study design, data collection and analysis, decision to publish, or preparation of the manuscript. The content is solely the responsibility of the authors and does not necessarily represent the official views of NIMH.

**Competing interests:** The authors have declared that no competing interests exist.

## Introduction

Anhedonia, the loss of interest in enjoyable activities or inability to experience pleasure, is one of the cardinal symptoms of depression and is frequently assessed as a trait construct [1]. One way to conceptualize anhedonia is by investigating recent changes in interest and pleasure, which has been found to be uniquely associated with depression and signal increased psychopathology [2]. This may be because a loss of pleasure indicates that a person formerly enjoyed and/or was interested in specific activities or social settings, but that those experiences are now less rewarding, and hence are not pursued or valued in the same manner [3,4]. Prior work has shown that anhedonia is related to decreases in reward-seeking behaviors, specifically the motivation to achieve or 'want' rewards (i.e., reward motivation) and decision-making for reward-related tasks or behaviors [5–9]. Thus, if one is unable to experience the pleasure or reinforcing nature of apparent rewards, it will likely result in further inhibition or avoidance of the reward in the future.

Persons with elevated anhedonia also often experience lower levels of positive affect, which can generally impact one's willingness or desire to achieve their goals [10–12]. As such, individuals suffering from anhedonia may struggle to motivate themselves to pursue formerly pleasurable stimuli, resulting in deficits related to reward motivation. This devaluative cycle of diminished positive affect may be the result of learned emotionally salient experiences. Specifically, Reward Devaluation Theory (i.e., RDT) posits that this reduced motivation for reward may result from repeated pairings of positivity with negative outcomes [4]. This learned association may lead to an avoidance of positivity and rewards, as some individuals with depression may experience negative affect in the face of positive stimuli. As such, depressed persons who devalue and avoid positivity likely also experience elevated anhedonia and reduced positive affect, perpetuating this cycle.

Given the detrimental impact anhedonia can have on personal functioning, is vital to identify factors that may be able to break this devaluative cycle and in turn likely increase reward motivation. One potential protective factor that may buffer against this diminished reward motivation seen in persons with elevated depression and anhedonia is action orientation, or the ability to upregulate positive affect in the face of stress or negative affect in order to pursue goals [13]. Whereas reward devaluation involves avoidance of positive affect due to fear that it will be coupled with negative outcomes, action orientation can be conceptualized as the theoretical inverse given that action orientation involves upregulating positive emotions (or the willingness to approach positivity, rather than avoid it) in the face of negative or stressful events [13,14]. Thus, action-oriented individuals may overcome obstacles in this manner, but individuals who are more state-oriented may find it difficult to do so [10]. In contrast to action orientation, state orientation can be conceptualized as an inability to upregulate positive affect, which can make it challenging for state-oriented individuals to make decisions and act efficiently [13]. Moreover, individuals who are state-oriented tend to dwell on their emotional states and ruminate over past events or failures, further limiting their ability to upregulate positive affect. As a result, state-oriented individuals may demonstrate impaired decision-making abilities, which can likely impact reward motivation and reward-seeking behaviors.

### Reward motivation in anhedonia

The Effort-Expenditure for Rewards Task (EEfRT) is a computer-based task that is particularly suitable to investigate the multifaceted reward decision-making process. It is frequently used to investigate human effort-based decision-making and emphasizes perseverance and cost-benefit analysis when making reward-related decisions [15]. In the EEfRT, participants

are presented with the option of engaging in a more difficult task with the prospect of gaining a larger reward (i.e., High Cost/High Reward (HC/HR)) or rather pursuing an easier task with a lower associated reward (i.e., Low Cost/Low Reward (LC/LR)). The beginning of each trial includes the probability that the accompanied trial will be a win trial, which factors into the cost-benefit analysis for the participant. In addition to this reward decision making process, participants must also engage in and complete the task for the opportunity to obtain the associated reward. Whereas the LC/LR task requires participants to complete 30 button presses with their dominant index finger, the HC/HR task requires 100 button presses with the nondominant pinky finger [15]. Thus, the EEfRT can obtain both measures of reward motivation and effortful behavior to achieve rewards by investigating individuals' decisions to complete easy or difficult tasks when considering their expended effort to gain the potential rewards. Compared to traditional self-report measures of anhedonia, the EEfRT can provide a more objective approach for measuring reward motivation [6,15,16]. Prior work indicates that depression and specifically, anhedonia, are associated with lower reward motivation behaviors when measured via the EEfRT [6,7,15,16]. Specifically, individuals with high levels of anhedonia are less likely to choose the task choice that requires a greater amount of effort (i.e., high effort) even with the possibility of a larger reward [7,15].

**The role of action orientation.** Prior work has further expanded the role of anhedonia on reward motivation and reward-seeking behaviors by exploring how the interactive effects of both anhedonia and action orientation influence reward motivation [17]. Specifically, when both anhedonia and action orientation were included in the model to predict reward motivation on the EEfRT, action orientation demonstrated an ability to buffer against anhedonia to some degree, likely by enhancing positive affect and goal pursuit, despite experiencing stress from a manipulation. Findings demonstrate that action orientation was linked to higher effort on the EEfRT at low levels of anhedonia; however, this buffering effect diminished at high levels of anhedonia, suggesting that an ability to upregulate positive affect and maintain motivation to achieve rewards (i.e., action orientation) may be impacted by elevated anhedonia. These findings highlight the important and complex interaction between action orientation and anhedonia, particularly in how these two factors can impact reward motivation on a behavioral task [17].

**The role of anxiety.** There is initial evidence that action orientation can impact the association between anhedonia and reward motivation on the EEfRT, but there is limited research regarding the impact of other emotional and clinical factors. For example, findings regarding the impact of anxiety on reward seeking behaviors remain mixed. Although anxiety has been associated with diminished reward motivation [18], other work examining the association between anxiety and reward-learning behaviors concluded that anxiety was not associated with diminished reward learning abilities [19,20]. In previous work utilizing a different effort-based experiment, unique motivational differences have been found between those with depression and anxiety based on potential rewards and losses [21]. This finding is partially supported by prior work that has investigated the role of anhedonia and anxiety on an emotional attention task (i.e., dot-probe task): anhedonia was associated with slower reaction times on positive valence trials, but was associated with faster reaction times when anxiety was elevated [22]. Another study concluded that social anxiety was not characterized by blunted positive affect [23]: individuals with social anxiety did not exhibit differential responsivity to reward via behavioral tasks or via neuroimaging, as there was no blunted activity in the ventral striatum, an area involved in reward processing. In fact, socially anxious individuals displayed increased brain activity in the default mode network during both reward anticipation and consummation. Thus, it is possible that reward motivation may not be diminished in those with social anxiety but may involve hypervigilance and fixation on

self-performance when in the presence of prospective reward. These findings suggest that anxiety may be able to as work as a buffer to increase reward behaviors, potentially due to the hypervigilance and arousal that commonly accompanies anxiety.

However, a substantial portion of the literature suggests that anxiety and anhedonia may actually have similar effects on reward motivation: they both diminish reward motivation and reward-seeking behaviors [18]. Research has shown that anhedonia and anxiety are closely tied with one another, as well as with future depression [24]. In particular, the relinquishment or avoidance of positivity is closely related to anxiety and may be a transdiagnostic mechanism between depression and anxiety [25]. Additionally, experimental studies have found that persons with elevated anxiety tend to engage in avoidance behaviors more than persons without anxiety, even when presented with the possibility of gaining larger rewards [26,27]. Given that avoidance is an important mechanism in anxiety, the devaluation of previously pleasurable activities may further perseverate avoidance of these activities due to anxiety, which can reinforce this cycle of avoidance between anxiety, anhedonia, and reward motivation [18].

As noted above, past research has demonstrated mixed findings regarding the impact of anxiety on reward motivation and its influence on the connection between anhedonia and reward motivation in particular. Specifically, it is unclear whether anxiety would either amplify or buffer the association between anhedonia and reward motivation. Thus, additional examination of the interactive effect of anhedonia and anxiety in relation to an experimental measure of reward motivation (i.e., the EEfRT) is needed to clarify the nature of these associations and provide potential implications for the treatment of anhedonia and reward devaluation.

## The current study

Anhedonia, action orientation, and anxiety may all uniquely impact reward motivation, with prior work implicating action orientation as a buffer [15,17,27]. One caveat to the prior work investigating the role of action orientation, however, was the utilization of a stress manipulation prior to completion of the EEfRT. This manipulation may have impacted the findings by facilitating a setting in which action orientation would be more applicable (i.e., in the face of stress) and thus amplified its ability as a buffer. However, given the potential utility of action orientation as a buffer against diminished reward seeking behaviors, additional effort to potentially replicate this finding without a stress manipulation would provide further implications of the role of action orientation. For example, if baseline action orientation, or the tendency to upregulate positive affect when facing challenges, can increase reward motivation even when stress in not concurrently being encountered, this would provide evidence for action orientation as a chronically accessible protective factor. Generalizing the role of action orientation as a buffering mechanism for reward motivation regardless of the situational context would provide support for targeting and building action orientation tendencies through future treatments. Therefore, the first aim of the study was to determine whether the findings of Bryant et al. [17] replicated in the absence of a stress manipulation. As such, we hypothesized that action orientation would act as a buffer for reward motivation, but only at low levels of anhedonia.

Moreover, given the mixed findings regarding the impact of anxiety, the second aim of this study was to investigate the impact of anxiety on the association between anhedonia and reward motivation on the EEfRT. As it remains unclear whether anxiety may further decrease the low reward motivation that is seen in anhedonia, or whether the hypervigilance and arousal that accompanies anxiety would act as a slight potential buffer, we investigated the interactive effect to assess these competing hypotheses. No specific hypotheses for the anxiety

model were outlined in the pre-registration, as this largely explorative examination sought to gain clarity with regards to these competing hypotheses. Specifically, we investigated the interactive effect of anhedonia and anxiety, in addition to the main effects, on reward motivation. This allowed us to more precisely examine whether anxiety amplifies and mimics the effects of anhedonia or whether anxiety acts as a buffer on reward motivation.

## Method

### Participants

One-hundred-one ($N$ = 101) participants ($M_{age}$ = 22.72, $SD_{age}$ = 8.68; 69.31% women; 54.46% White) with a range of depressive symptoms were recruited to complete six weekly sessions of a larger research study, as well as a follow-up session that took place six weeks later. Participants were recruited via ads posted in hospitals and outpatient clinics, community businesses, and at a large Southern university. Recruitment for the study began on 1/28/2015 and ended on 4/26/2017. Individuals were first directed to an online screening process on Qualtrics, which included the Quick Inventory of Depressive Symptomatology (QIDS-SR; [28]). To create a normal distribution of pre-screen depression scores, we aimed to recruit an equal number of participants in the low, moderate, and severe ranges. The sample consisted of 47 participants with mild depression symptoms, 24 participants with moderate depression symptoms, and 30 participants with severe depressive symptoms. The mean of the QIDS-SR pre-selection scores fell within the moderate range of depressive symptoms ($M$ = 11.55, $SD$ = 5.63; [28]). The current study received approval from the university's Institutional Review Board (IRB #14-242). All participants provided written consent before participating in the study.

### Materials

**Specific loss of interest and pleasure scale (SLIPS).** The Specific Loss of Interest and Pleasure Scale (SLIPS; [2]) is a 23-item self-report measure that assesses recent changes in the anhedonia, particularly to social experiences, over the past two weeks. The SLIPS can capture both recent changes and trait anhedonia, with all items scored on a 4-point Likert scale ranging from 0-3, and responses of "3" are recoded to "0" given that they reflect trait anhedonia, rather than recent changes (e.g., "I have *never* enjoyed leisure activities that involve other people"). Thus, sum scores range from 0-46, with higher scores corresponding to higher anhedonia severity. The SLIPS demonstrated good internal consistency in the current study ($\alpha$ = .91)

**Beck anxiety inventory (BAI).** The Beck Anxiety Inventory (BAI; [29]) is a 21-item self-report measure that examines anxiety symptomatology. The BAI items are scored on a 4-point Likert scale with responses ranging from 0 ("not at all") to 3 ("severely"), where higher scores represent more severe anxiety symptoms. Total BAI scores represent a range of anxiety symptoms from minimal to severe levels of anxiety. The BAI demonstrated excellent internal reliability in the current study ($\alpha$ = .92).

**Action control scale (ACS).** The Action Control Scale (ACS; [13]) is a 36-item self-report scale that assesses for action versus state orientation. Statements are presented with two possible answer choices that reflect either an action- or state-orientated behavior. The ACS consists of three subscales: failure-related action orientation versus preoccupation (AOF), decision-related action orientation versus hesitation (AOD), and successful performance-related action orientation versus volatility (AOP). Consistent with Bryant et al. [17], only the AOD subscale was included in the current analyses, as it reflects the ability to upregulate positive affect in the face of negative or stressful information [10,13]. Scores range from

0-12 for each subscale, with higher scores indicating more action-oriented values. The ACS demonstrated good internal consistency in the current study (α = .86).

**The effort-expenditure for rewards task (EEfRT).** The Effort-Expenditure for Rewards Task (EEfRT; [15]) is a behavioral task used to assess reward motivation. The EEfRT quantifies the amount of effort one is willing to exude in response to a reward and measures the extent to which individuals will expend greater effort or work harder in the hopes of receiving a greater reward (i.e., high cost/high reward). The paradigm uniquely varies the probability of receiving an award upon completion of the task, as well as the amount or size of reward, and uses two different task choices of varying difficulty (e.g., low cost/low reward (LC/LR) versus high cost/high reward (HC/HR)). Replicating the procedures of Bryant et al. [17], the easy task (i.e., LC/LR) required participants to complete 30 button presses with their dominant index finger within seven seconds in order to be eligible for potential reward. Participants were only eligible to win a set amount ($1.00) on each of the easy trials, and they were not able to win any prize amounts larger than $1.00 on these easy trials. In contrast, the difficult task (i.e., HC/HR) required more effort but included a larger reward upside: participants needed to complete 100 button presses with the pinky finger of their non-dominant hand within 30 seconds in order to be eligible to win a greater reward. Participants who successfully completed the HC/HR trials were eligible to win a larger sum of money, as the potential winnings varied from $1.24 to $4.30. All participants were video recorded while completing the EEfRT, and both undergraduate research assistants and a graduate research assistant monitored the live video during the task to ensure compliance with task instructions.

It is important to note that merely completing one of the tasks was not an assurance of compensation regardless of their difficulty level, as certain trials were pre-determined to be "win" trials and others were "no win" trials. Upon completion of the entire paradigm, two "win" trials were selected at random, and participants received the amount of money that they won on those trials. Although participants were not made aware of which trials would be the two "win" trials, participants were informed of the probability of each trial being a "win" trial before choosing the LC/LR or HC/HR option for each trial. The probability ranges included high (88%), medium (50%), and low (12%), and informed participants the probability that the upcoming trial would be a "win" trial if successfully completed. Thus, participants knew the likelihood of receiving the reward should they complete the task and therefore made decisions regarding the amount of effort they were willing to exude on each trial. As such, the knowledge of the probability (i.e., low to high) that their effort would be met with reward provides important information into their motivation to pursue and potentially achieve reward.

Of importance, there were a nearly equal number of high, medium, and low probability trials throughout the study. Participants were given 20 minutes to complete as many trials as possible; thus, the number of trials varied across participants. If participants did not choose the task within five seconds, one of the trials was randomly assigned to them. Trials were counterbalanced and presented in pseudorandomized order. The trials were fixed in that each participant received the same trials in the same order, but some participants received more trials depending on how quickly they completed each trial. Nevertheless, all participants received approximately an equal number of each trial probability due to counterbalancing, regardless of how many trials were completed. The smallest number of trials completed by a participant was 52 (18 of 12%; 17 of 50%; and 17 of 88% probability trials). The largest number of trials completed by a participant was 186 (61 of 12%, 63 of 50%, and 62 of 88% probability trials). At the end of the task, the experimenter noted the total reward from the two randomly selected win trials on the EEfRT that was shown on the screen, which was paid to participants at the end of the session.

## Procedure

Participants ($N$ = 101) enrolled in the study attended six weekly in-person sessions and could participate in a follow-up session approximately six weeks later. All participants attended a minimum of four of the six weekly sessions, and the follow-up session was optional. During the first session, participants completed an in-person consent process with a graduate student, the EEfRT, SLIPS, and BAI, as well as a variety of other tasks and measures that were part of a larger study [30]. The ACS was only administered at the follow-up session. In addition to the money earned on the EEfRT, participants were also paid $20 for each session for the first five weeks (sessions 1–5), $50 in week 6, and $50 at the follow-up session. At the end of each session, regardless of their responses on the self-report measures or tasks, local and national resources for mental health services were provided to each of the participants.

One participant was not included in the current analyses due to missing EEfRT data, resulting in adequate data for 100 participants from the first weekly session for the current analyses. Moreover, the ACS was only assessed at the follow-up session, which was optional and thus included valid and complete responses from 75 participants (note there is a typo regarding the expected sample size in the pre-registration document). Thus, in an effort to use all available data to maximize power and remain consistent with using the same EEfRT data in analyses across models, the EEfRT, anhedonia, and anxiety measures were all completed during the first session, whereas only action orientation was obtained at the follow-up session. This resulted in 100 and 75 complete observations in the anxiety and action orientation models, respectively. One of the main goals of the study was to examine whether prior findings from Bryant et al. [17] would replicate using a nearly identical procedure but with the absence of a stress manipulation. Thus, since the current study included a sample size nearly identical to that used by Bryant and colleagues and slightly larger than another study that examined anhedonia in relation to the EEfRT using GEE modeling [15], we anticipated that all analyses would be adequately powered.

## Data analytic plan

All hypotheses and the data analytic plan were pre-registered via AsPredicted.org (#105462) before analyses were conducted. The pre-registration also included potential analyses with several types of anhedonia measures, but it was determined to be outside the scope of the current investigation given the study aims. The analyses were conducted in SPSS Version 29 using a generalized estimating equation (GEE) model, following prior work that has utilized GEE modeling given that the EEfRT includes examination of data that are interdependent with an unknown level of correlation with one another [15,17]. The outcome variable was a dichotomous variable reflecting either the HC/HR or LC/LR task choice; thus, a binary logistic regression was used. The primary GEE model included reward amount, probability, and expected value (i.e., probability x reward amount) as predictor variables, as was done in prior work [17]. In the first model, anhedonia and action orientation were entered into the model as continuous predictor variables and an interaction term was created.

As outlined in the pre-registration, the planned second GEE model included anxiety as a covariate, rather than action orientation. However, because of multicollinearity concerns due to the high amount of shared variance between anhedonia and anxiety (>=.59), we were unable to run the analysis as a GEE model. Therefore, we instead stipulated the second model using moderation analyses, deviating from our pre-registered data analytic plan. Moderation analyses were conducted using the PROCESS Model 1, utilizing a 5,000-sample bootstrapping technique with 95% confidence intervals.

## Results

### Data cleaning

We first examined responses on all outcomes via boxplots and frequency analyses. Following recommendations by Tabachnick and Fidell [31], cases with z-scores> |3.29| were identified as outliers and recoded as +/-3.29. The only outlier in the current dataset included a BAI z-score of 3.41, which was recoded to 3.29. The distributions of all other variables were within normal limits (skewness < 3.0, kurtosis < 10.0; [32]). See Table 1 for descriptive statistics of the measures used. In addition, on 1.1% of the EEfRT trials, participants did not choose the LC/LR or HC/HR task option before the decision time expired and thus, were randomly assigned to one of the tasks. These trials that lacked a participant choice were removed prior to running analyses.

### Model 1

The first model examined the main effects of anhedonia and action orientation on reward motivation, as well the anhedonia × action orientation interaction. The probability of win trials ($b = 1.96$, $p = .020$), reward amount ($b = 0.37$, $p = .048$), and expected value ($b = 0.56$, $p = .040$) were significantly associated with higher HC/HR choices. Higher probability of being eligible for reward and larger potential winnings were related to a greater likelihood of choosing the HC/HR task. Additionally, anhedonia ($b = −0.17$, $p < .001$) was significantly associated with task choice: higher levels of anhedonia were associated with lower reward motivation (i.e., less likely to choose the HC/HR option). Contrary to our hypotheses, action orientation exhibited a negative association with reward motivation ($b = −0.38$, $p < .001$). However, the interaction between anhedonia x action orientation was significant ($b = 0.02$, $p < .001$).

We followed up these findings with an exploratory moderation analysis in PROCESS Model 1 to further probe the anhedonia x action orientation interaction. Identical to the GEE model, the 75 available observations with complete data on the ACS were utilized to ensure complete data in the moderation analyses. We included anhedonia as the predictor, action orientation as the moderator, and task choice remained the outcome. The moderation analyses not only corroborated the findings initially outlined using our pre-registered GEE modeling but also provided additional evidence for the significant influence of anhedonia and action orientation on reward motivation. The moderation analysis revealed that anhedonia was again negatively associated with reward motivation ($b = −0.04$, $p < .001$), but action orientation was no longer significantly related to reward motivation ($b = −0.02$, $p = .087$). However, the anhedonia x action orientation interaction ($b = 0.01$, $p < .001$) was significantly associated with task choice. We followed up this significant interaction with a trend analysis and results indicated that anhedonia was strongly and negatively associated with reward motivation at low levels (i.e., −1 SD) of action orientation, ($b = −0.06$, $p < .001$), and anhedonia demonstrated an attenuated, but negative, association with reward motivation at high levels (i.e., +1 SD) of action orientation ($b = −.02$, $p < .001$). Thus, these

**Table 1. Descriptive Statistics.**

| Name | Mean | Standard deviation | Skewness | Kurtosis |
|---|---|---|---|---|
| SLIPS | 10.06 | 8.22 | 1.03 | 0.48 |
| AOD | 6.69 | 3.25 | −0.37 | 0.55 |
| BAI | 14.20 | 10.76 | 1.18 | 0.48 |

*Note.* SLIPS = anhedonia; AOD = action orientation; BAI = anxiety.

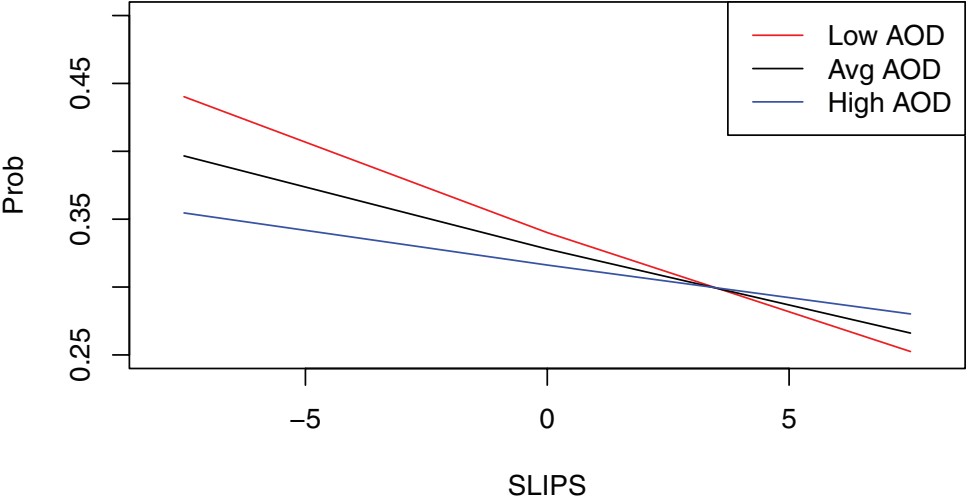

**Fig 1. Interaction of Anhedonia and Action Orientation.** *Note*. This figure displays the moderation graph from the PROCESS analysis. The figure shows the relationship between anhedonia and choice probability at low, average, and high levels of action orientation. Prob= probability of choosing the HC/HR task; SLIPS= anhedonia; AOD = action orientation. Red line = low action orientation, black line = average action orientation, blue line = high action orientation.

moderation results indicate that action orientation does buffer against anhedonia in its association with reward motivation, providing a partial replication of the findings from Bryant et al. [17]. In the prior work, the buffering effect was found only when anhedonia was low. In contrast within the current sample, this buffering effect only emerged when anhedonia levels were high (see Fig 1).

## Model 2

As noted above, due to the multicollinearity of the predictor variables in the second model, namely between anhedonia and anxiety, GEE was not an appropriate model for the data and thus could not be run. Therefore, as was done in the exploratory analyses for the first model, we conducted a moderation analysis using PROCESS Model 1 to investigate the effect of anhedonia and anxiety on reward motivation. All available 100 observations were utilized in Model 2 to maximize power. Anhedonia was again entered as the predictor variable and task choice as the dependent variable (as done in Model 1). However, anxiety was included as the moderator, rather than action orientation. This second moderation analysis revealed that anhedonia was again significantly associated with a lower probability of choosing the HC/HR task ($b = -0.02$, $p < .001$). However, neither anxiety ($b < 0.01$, $p = .287$) nor the anhedonia x anxiety interaction ($b < 0.01$, $p = .121$) were significant in this model, evidencing a divergent pattern from our first model that included action orientation. Although anhedonia and anxiety did not emerge as a significant interaction in relation to reward motivation, the graphical representation of the data at high (i.e., +1 SD) and low (i.e., −1 SD) levels of anhedonia and anxiety displays a trend, albeit not significant (see Fig 2).

## Discussion

The current study sought to investigate how clinical factors, namely action orientation and anxiety, influence the relationship between anhedonia and reward motivation. Anhedonia

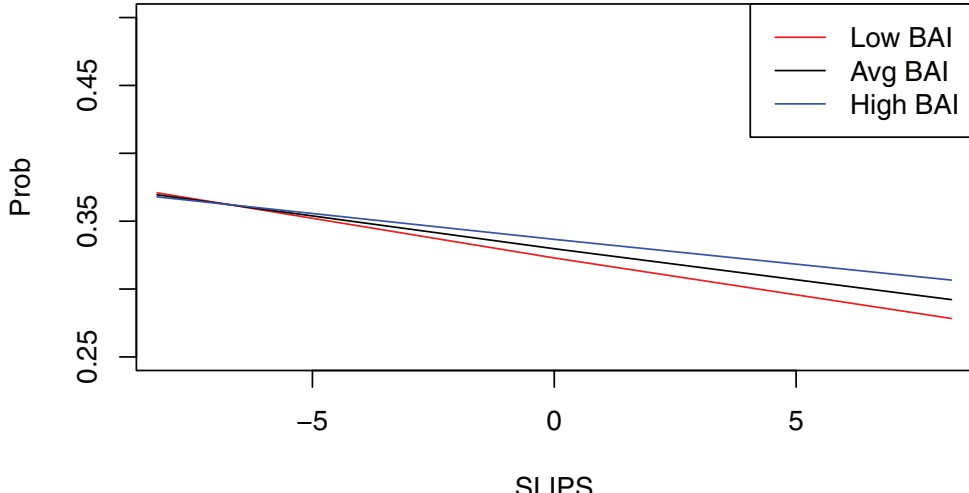

**Fig 2. Interaction of Anhedonia and Anxiety.** *Note*. This figure displays the moderation graph from the PROCESS analysis. The figure shows the relationship between anhedonia and choice probability at low, average, and high levels of anxiety. Prob= probability of choosing the HC/HR task; SLIPS= anhedonia; BAI= anxiety. Red line = low anxiety, black line = average anxiety, blue line = high anxiety.

has consistently been associated with low reward motivation via the EEfRT task [7,15,17]. However, the precise impact of additional factors, such as action orientation and anxiety, on this relationship remained unclear. Therefore, the first aim of the current study was to examine whether the findings of Bryant et al. [17], which showed that action orientation acted as a buffer for reward motivation at various levels of anhedonia, replicated in the absence of a stress manipulation. We hypothesized that action orientation would act as a buffer for reward motivation, but only at low levels of anhedonia. The second aim of the study was to further examine the role of anxiety as a potential amplifier or buffer between anhedonia and diminished reward motivation. We further describe the findings of our two models below.

## The impact of anhedonia and action orientation on reward motivation

Our results revealed that anhedonia had a significant, negative impact on reward motivation. This is congruent with Treadway and Zald's [33] concept of "decisional anhedonia," in which the authors characterize a decreased ability for normative decision-making as one manifestation of anhedonia symptoms. These results are also consistent with Reward Devaluation Theory (RDT), which indicates that certain people with depressive symptoms may devalue or avoid rewarding experiences [4]. As a result, it stands to reason that anhedonia would be linked to task choice in this study. There may also be an avoidance or undervaluation of social rewarding experiences, as anhedonia is often a barrier to enjoyment of social contexts [34]. These individuals who do not value rewarding stimuli, possibly as a result of negative experiences in the past, may be less inclined to pursue potentially rewarding activities [4].

Further, our results provided partial evidence for our hypothesis that action orientation would act as a buffer between anhedonia and reward motivation. At low levels of action orientation, the negative association between anhedonia and reward motivation remained; however, this impact of anhedonia was attenuated when action orientation was high. Therefore, persons with greater action orientation may be more likely to upregulate positive affect and motivation in the pursuit of reward, even if they experience high levels of anhedonia. This partially supports our hypothesis, as action orientation did act as a buffer of reward

motivation, but only when anhedonia was high, as this same pattern was not seen when anhedonia was low. These findings support prior work and suggest that anhedonia generally impacts reward motivation, but action orientation may be able to act as a buffer against this negative relationship [10].

The findings regarding action orientation provide evidence for a partial replication for the results of the Bryant et al. [17] study: they discovered an interaction between anhedonia and action orientation, where action orientation served as a buffer against diminished reward motivation at low levels of anhedonia, but at high levels, anhedonia debilitated action orientation, making it no longer a promotive factor. However, the current results indicate that when anhedonia is low, reward motivation is high in general regardless of action orientation. One potential explanation for these slightly different findings is that the current study consisted of a community sample that was recruited for a varying range of depressive symptoms from none to severe, thus likely impacting anhedonia severity as opposed to the student sample included in the study by Bryant and colleagues. Another potential explanation for the findings is that the current study did not implement a stress manipulation; thus, the ability to upregulate positive affect in the face of stress (i.e., action orientation) may have expanded in its ability to be a promotive factor at high levels of anhedonia given there was likely lower levels of stress during the experiment.

## The impact of anhedonia and anxiety on reward motivation

The second aim of the current study was to investigate the combined effect of anhedonia and anxiety in relation to reward motivation. Past research has established a connection between anhedonia and anxiety [18,22,24]. However, the nature of this relationship, specifically whether there is either a reinforcing/amplifying or a buffering effect in relation to reward motivation, remained unclear. Therefore, our second model examined the effects of anhedonia and anxiety on reward motivation through the EEfRT.

The clear negative relationship between anhedonia and reward motivation remained even when anxiety was included in the model. Neither the main effect of anxiety nor the interaction of anhedonia x anxiety was significantly related to reward motivation. However, examination at high and low levels of anhedonia and anxiety displayed a potential weak and non-significant buffering effect of anxiety that should be further explored in future work. Specifically, a pattern emerged only when anhedonia was high, in which those with high levels of anxiety exhibited greater reward motivation than those with average and low levels of anxiety, although this interaction was not statistically significant. This is partially supported by past work which found differential responsivity to rewards and losses in depressed and anxious groups [21]. Although they utilized a different task, individuals with anxiety displayed greater motivation to avoid potential losses than those with depression. Additionally, anxiety was associated with greater extrinsic motivation, compared to the depressed group, during the reward task. The motivation to avoid loss (i.e., the opportunity to gain reward) in the EEfRT paradigm, as well as being motivated by the monetary extrinsic reward in the EEfRT task may be potential factors that contributed to the current findings. Recent work using advanced learning algorithms found that depending on the learning strategy employed, differences in reward decision making emerges based on anhedonia and anxiety severity [35]. That is, differences in reward decision making can emerge depending on if the algorithm considers all previous trials and probabilities or just the trial that immediately preceded.

The findings regarding the interactive effect of anhedonia and anxiety require additional research to further clarify their role in relation to reward motivation. Another possible explanation of the current finding is that there is a large amount of overlap between anhedonia and

anxiety, which could account for the lack of statistically significant findings in the moderation model. This explanation is consistent with recent theoretical work by Taylor et al. [18], which suggests that self-reported anxiety and anhedonia may share partial pathways that predict reward deficits. However, a weak buffering effect does appear to be present. Moreover, past research has shown that anxiety and depression are linked with one another, and appear to be mediated by avoidance [36]. Given that anticipatory avoidance is a key factor in anhedonia [37], it may be that a relationship between anxiety and reward motivation (i.e., a common feature in depression) only emerges when anhedonia or avoidance of prospective reward is high. Future research should continue to include anxiety, anhedonia, and personality components related to the upregulation of positivity to better understand the interplay among these variables.

## Clinical implications

Our findings suggest that action orientation, or the upregulation of positive affect in the face of negativity, may be a useful construct to introduce in clinical settings because it can act as a buffer against anhedonia and possibly increase reward seeking behaviors. There are several emerging treatments that may specifically be beneficial in targeting anhedonia symptoms [38]. Individuals who experience elevated anhedonia or reduced motivation to seek rewarding/pleasurable experiences may benefit from therapies that focus on upregulating the positive valence systems [39–41], with the caveat that persons who devalue or avoid positivity may not choose to engage in these treatments [42]. These treatments focus on upregulating positive affect through various skills such as scheduling pleasurable activities and implementing positivity savoring techniques.

## Strengths/limitations

One strength of the current study was the inclusion of a behavioral task to quantify reward motivation. While self-report measures are valuable, they can be subject to demand characteristics. Moreover, the EEfRT task requires participants to not only make a decision regarding the amount of effort they are willing to expend for a potential reward, but also requires them to engage in the effortful behavior to earn the reward. Thus, the EEfRT provides a more objective measure of reward motivation and produces a range of unique datapoints (e.g., reward amount) that are difficult to capture with self-report measures [6,15,16]. A related advance that measures the tendency to devalue reward but has increased ecological validity is the newly introduced Valence Selection Task (VST; [43]). Future research examining the interface of willingness to expend effort and reward devaluation can incorporate both the EEfRT and the VST.

One of the limitations of the current study is the lack of a clinical sample. Although participants were pre-selected based on depression scores to ensure a spread of symptom severity, we did include persons with both healthy ranges and psychopathology ranges. Thus, although a clinical sample was not used in the current study, the current sample consisted of a spread of depression symptoms, including several participants with severe depressive symptoms. This methodology can provide a more comprehensive, dimensional view of psychopathology, rather than just relying on clinical diagnoses. As such, examining reward motivation in a sample with a varying spread of depression symptoms provided valuable insight into how anhedonia affects reward motivation.

Another limitation of this study is the smaller sample size in the action orientation analyses. Since the ACS was only administered at the follow up session, there was naturally attrition that resulted in a smaller sample size than the initial session. Thus, while the dataset consisted of a larger

sample for anhedonia, anxiety, and reward motivation, the study was limited in its sample of action orientation. However, the sample size was nearly identical to a past study that examined action orientation in relation to reward motivation [17]. Future research would benefit from further examining the role of action orientation on reward motivation in a larger, more robust, sample.

## Conclusion

Anhedonia has continually been shown to negatively influence reward motivation [5,9,17]. If individuals with anhedonia, or depression more generally, do not seek out rewarding activities due to a lack of motivation, this may further reinforce this avoidance process; thus, further devaluing what would seem to be an enjoyable experience. This study used an experimental paradigm to further investigate the role of action orientation to counteract or act as a buffer for this diminishment of reward motivation without a stress manipulation. The results suggest that while anhedonia is extremely influential in limiting reward motivation behaviors, high levels of action orientation help to independently reduce the negative influence of anhedonia. As such, action orientation may be a protective factor that can allow persons with elevated anhedonia to overcome their low positive affect and engage in reward-seeking behaviors. This suggests that learning to upregulate positive emotions in the face of negativity may act as an effective strategy to potentially improve reward motivation.

## Acknowledgements

None

## Author contributions

**Conceptualization:** Michael R. Gallagher, E. Samuel Winer.

**Data curation:** Amanda C. Collins.

**Formal analysis:** Michael R. Gallagher, Samantha L. Anduze, E. Samuel Winer.

**Funding acquisition:** E. Samuel Winer.

**Investigation:** Amanda C. Collins, E. Samuel Winer.

**Methodology:** Michael R. Gallagher, E. Samuel Winer.

**Supervision:** E. Samuel Winer.

**Visualization:** Michael R. Gallagher.

**Writing – original draft:** Michael R. Gallagher, Amanda C. Collins, Samantha L. Anduze.

**Writing – review & editing:** Michael R. Gallagher, Amanda C. Collins, Samantha L. Anduze, E. Samuel Winer.

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
