## [Decision Letter · Decision Letter 0]

29 Dec 2024

PONE-D-24-40263An experimental investigation of the role of anhedonia, anxiety, and action orientation on reward motivationPLOS ONE

Dear Dr. Gallagher,

Thank you for submitting your manuscript to PLOS ONE. After careful consideration, we feel that it has merit but does not fully meet PLOS ONE’s publication criteria as it currently stands. Therefore, we invite you to submit a revised version of the manuscript that addresses the points raised during the review process. Both myself, and an expert in the field, have found your study and manuscript to be valuable contributions to the literature. We both point to the inclusion of a little more breadth concerning the two roles of anxiety in reward motivation in a revised version. In addition, when revising your text I urge you to consider the additional, excellent comments provided by Reviewer 1 as these will improve the universal impact of your piece. 

We look forward to receiving your revised manuscript.

Kind regards,

Herb Covington, Ph.D.

Academic Editor

PLOS ONE

Journal requirements: When submitting your revision, we need you to address these additional requirements. 1. Please ensure that your manuscript meets PLOS ONE's style requirements, including those for file naming. The PLOS ONE style templates can be found at https://journals.plos.org/plosone/s/file?id=wjVg/PLOSOne_formatting_sample_main_body.pdf and https://journals.plos.org/plosone/s/file?id=ba62/PLOSOne_formatting_sample_title_authors_affiliations.pdf. 2. Please include a caption for figure 1 and 2.  3. Thank you for stating the following financial disclosure:  [Research reported in this publication was supported by the National Institute of Mental Health (NIMH) under Award Number R15MH101573 (PI: ESW). The content is solely the responsibility of the authors and does not necessarily represent the official views of NIMH].  Please state what role the funders took in the study.  If the funders had no role, please state: ""The funders had no role in study design, data collection and analysis, decision to publish, or preparation of the manuscript."" If this statement is not correct you must amend it as needed. Please include this amended Role of Funder statement in your cover letter; we will change the online submission form on your behalf. 4. When completing the data availability statement of the submission form, you indicated that you will make your data available on acceptance. We strongly recommend all authors decide on a data sharing plan before acceptance, as the process can be lengthy and hold up publication timelines. Please note that, though access restrictions are acceptable now, your entire data will need to be made freely accessible if your manuscript is accepted for publication. This policy applies to all data except where public deposition would breach compliance with the protocol approved by your research ethics board. If you are unable to adhere to our open data policy, please kindly revise your statement to explain your reasoning and we will seek the editor's input on an exemption. Please be assured that, once you have provided your new statement, the assessment of your exemption will not hold up the peer review process.

Reviewers' comments:

Reviewer's Responses to Questions

**Comments to the Author**

1. Is the manuscript technically sound, and do the data support the conclusions?

Reviewer #1: Partly

Reviewer #2: Yes

2. Has the statistical analysis been performed appropriately and rigorously? 

Reviewer #1: Yes

Reviewer #2: Yes

3. Have the authors made all data underlying the findings in their manuscript fully available?

Reviewer #1: Yes

Reviewer #2: Yes

4. Is the manuscript presented in an intelligible fashion and written in standard English?

Reviewer #1: No

Reviewer #2: Yes

5. Review Comments to the Author

Reviewer #1: Gallagher et. al. investigate how action orientation and anxiety moderate the association between anhedonia and reward motivation, as measured by the EEfRT task. The study aims to test two hypotheses: (1) whether, in the absence of stress manipulation, action orientation mitigates the decrease in reward motivation at low levels of anhedonia, and (2) whether the effect of anhedonia on reward motivation depends on the level of anxiety.

The manuscript addresses an important and complex topic, employing computational modeling to examine nuanced relationships between key variables. The finding that action orientation buffers diminished reward motivation at high levels of anhedonia is particularly intriguing and provides valuable insights into individual differences in reward-related behaviors. However, several aspects of the manuscript could benefit from refinement to enhance clarity, organization, and coherence. Below, I outline major and minor points for consideration.

Major Points

1. Readability and Clarity

The manuscript could benefit significantly from improved readability. Many sentences are excessively long, with numerous commas and parentheses, making the text difficult to follow. Additionally, some sentences contain redundant information that could be omitted without altering their meaning. Streamlining these sentences would improve flow and accessibility.

For instance:

Lines 7-14 of the introduction describe the relationship between anhedonia, reward-seeking behavior, and reward motivation. The first sentence introduces the concept, while the second and third sentences reiterate the same ideas with minor variations. This redundancy detracts from the flow and clarity of the argument.

Similarly, lines 15-21 of the introduction (continued on the first two lines of page 4) describe a cycle in which anhedonia leads to reduced positive affect and motivation, which in turn reinforces anhedonia. While this cycle is important, the text does not explain what might initiate or perpetuate it, leaving the reader with unanswered questions.

I recommend revising these sections for conciseness and coherence.

2. Organization of the Introduction

The introduction feels disorganized, with weak logical connections between sentences and vague descriptions of key constructs. For instance:

The section on action orientation introduces the term without providing a clear, concise definition. Sentences such as "One potential protective factor..." and "action orientation can be conceptualized as the theoretical inverse of reduced reward motivation..." are verbose and repetitive. Furthermore, the cited source does not define action orientation as the inverse of diminished reward motivation but instead focuses on volitional control and emotional regulation. Clarifying these points and ensuring alignment with cited sources would strengthen the introduction.

Additionally, the rationale for removing stress manipulation is not adequately explained, especially given that action orientation is described as a response to stress. A clearer explanation of the study's design choices would help readers understand the context and importance of the findings.

3. Role of Anxiety

The section discussing anxiety's role in reward motivation is somewhat ambiguous. At times, the text suggests that anxiety might buffer reward behaviors (e.g., "anxiety may work as a buffer to increase reward behaviors"), while at other points, it posits that anxiety and anhedonia have similar effects in diminishing reward motivation. These conflicting conclusions are not reconciled, leaving the reader uncertain about the primary hypothesis.

To address this, I suggest:

Providing a theoretical framework that explains why anxiety might have different effects in different contexts.

Clarifying the hypotheses and presenting evidence more cohesively.

Including specific, testable hypotheses for future research to resolve these mixed findings.

4. Figures and Visual Presentation

The figures are low in quality, and the colors used are difficult to distinguish between conditions. I recommend improving the resolution of the figures and selecting colors that are more discernible for readers.

5. Missing Data

The manuscript states that 25 participants did not complete the action orientation scale, but it is unclear how this missing data was managed in the moderation analysis. Since these models require complete data for the moderator, predictor, and outcome variables, it seems likely that the sample size was effectively reduced to 75. However, the manuscript implies that all 100 observations were used. Please clarify how missing data were handled, discuss any potential issues arising from the unevenness, and explain how these issues were addressed.

Minor Points

1- Title: The title reflects the key variables and experimental approach but could be refined to emphasize the computational modeling aspect and the interaction between variables. This would make it clearer and more informative.

2- Participant Compensation: There is inconsistency in describing participant compensation. Page 10, lines 20-22 state that participants were paid based on two randomly selected win trials, while page 11, lines 11-13 suggest they were paid the total of all wins. Please clarify which description is accurate. Furthermore, it is stated that at the end of each session participants were provided with local and national referral resources (page 11, last two lines), but no mention of what kind of resources they were referred to.

3- EEfRT Task Explanation: Page 5, lines 21-22 describe how reward motivation and effortful behavior were derived from the EEfRT task but do so vaguely. Explicitly defining how these measures were calculated (e.g., low-probability rewards for reward motivation, HC/HR trials for effortful behavior) would eliminate room for speculation.

4- Compliance with Task Instructions: Page 10, line 16 mentions task instructions (e.g., using the pinky of the non-dominant hand). Were participants supervised, or were video recordings used to ensure compliance?

5- Trial Distribution: Page 11, line 10 states that the number of trials was not fixed as participants had to complete as many trials as they could in a 20 minute timespan. How, then, did the authors ensure equal numbers of high, medium, and low-probability trials?

6- Inclusion Criteria: Page 11, line 18 mentions that only participants who attended at least four sessions were included, yet only data from the first session were used in the main analysis. Clarifying the rationale for this criterion would improve transparency.

8- Grammar and Typos: There are minor grammatical errors (e.g., page 3, lines 3-4, where the verb does not agree with the subject) and typos (e.g., page 6, line 7). A careful proofreading would address these issues.

Conclusion

Gallagher et al.'s study tackles a critical topic in the field, and the findings offer valuable insights into the moderating role of action orientation. However, addressing the issues outlined above—especially those related to clarity, organization, and the handling of missing data—will enhance the manuscript's overall impact. I hope these suggestions help improve the clarity and accessibility of the study for readers.

Reviewer #2: This manuscript, crafted by Gallagher and colleagues, provides an insightful evaluation of the psychological constructs ‘action orientation’ and ‘anxiety’ and their interactions with anhedonia, when modulating reward motivation in the EEfRT task. The two aims of this study were aimed at exploring: 1. that lower intensities of action orientation reflect decreases in reward motivation when anhedonia is present, and 2. that anxiety, depending on intensity, increases or decreases reward motivation in conjunction with anhedonia.

The present analyses derived from a solid sample of individuals with depressive symptoms reveals that 'action orientation' can protect ‘reward motivation’ when anhedonia is also highly expressed. While the results of the moderation analysis for anhedonia and anxiety on reward motivation were less robust, I did find that the ability of high anxiety to lean-towards-a-buffering-effect to be an important contribution for many disciplines that are concerned with anhedonia, affective states, or reward processing. My only comment for the authors to consider here is that a little more science pertaining to 'anxiety' and it ability to modulate ‘reward motivation’ be provided in the Discussion section.

6. PLOS authors have the option to publish the peer review history of their article (what does this mean?). If published, this will include your full peer review and any attached files.

Reviewer #1: No

Reviewer #2: **Yes: **Herb Covington

---

## [Author Response · Author response to Decision Letter 1]

11 Feb 2025

Thank you for the opportunity to revise and resubmit our manuscript to PLOS One. Please see our response to reviewers document for point-by-point responses to all suggested revisions.

---

## [Editor Report · Decision Letter 1]

13 Feb 2025

When effort pays off: An experimental investigation into action orientation and anxiety as buffering factors between anhedonia and reward motivation

PONE-D-24-40263R1

Dear Dr. Gallagher,

We’re pleased to inform you that your manuscript has been judged scientifically suitable for publication and will be formally accepted for publication once it meets all outstanding technical requirements.

Kind regards,

Herb Covington, Ph.D.

Academic Editor

PLOS ONE
---

## [Editor Report · Acceptance letter]

PONE-D-24-40263R1

PLOS ONE

Dear Dr. Gallagher,

I'm pleased to inform you that your manuscript has been deemed suitable for publication in PLOS ONE. Congratulations! Your manuscript is now being handed over to our production team.

Kind regards,

on behalf of

Dr. Herb Covington

Academic Editor

PLOS ONE